# Limited Effects of Class II Transactivator-Based Immunotherapy in Murine and Human Glioblastoma

**DOI:** 10.3390/cancers16010193

**Published:** 2023-12-30

**Authors:** A. Katherine Tan, Aurelie Henry, Nicolas Goffart, Sofie van Logtestijn, Vincent Bours, Elly M. Hol, Pierre A. Robe

**Affiliations:** 1Department of Translational Neuroscience, University Medical Center Utrecht (UMCU) Brain Center, Utrecht University, 3584 CX Utrecht, The Netherlands; a.k.tan@umcutrecht.nl (A.K.T.); e.m.hol-2@umcutrecht.nl (E.M.H.); 2Department of Human Genetics, University of Liège, 4000 Liège, Belgium; 3Department of Neurosurgery, University Medical Center Utrecht (UMCU) Brain Center, Utrecht University, 3584 CX Utrecht, The Netherlands

**Keywords:** glioblastoma, class II transactivator, tumor microenvironment, tumor-associated macrophages, tumor-infiltrating lymphocytes, natural killer cells

## Abstract

**Simple Summary:**

The major histocompatibility complex type II is downregulated in glioblastoma (GB) due to the silencing of the major transcriptional regulator class II transactivator (CIITA). Recent studies have posed CIITA as a possible target in treating patients with GB. However, our results show that in mice, vaccination experiments with wildtype (-WT) and CIITA-expressing cells (-CIITA) equivalently protect against subsequent intracerebral challenges with GL261 cells. Adoptive cell transfer experiments likewise showed a similar antitumor potential of lymphocytes harvested from mice implanted intracerebrally with GL261-WT or -CIITA. Human GB-infiltrating myeloid cells, lymphocytes and NK cells remained mostly inactivated and/or tumor-supportive when in co-culture with GB cells, regardless of CIITA. Altogether, these results question the therapeutic potential of CIITA-mediated immunotherapy in glioblastoma.

**Abstract:**

Background: The major histocompatibility complex type II is downregulated in glioblastoma (GB) due to the silencing of the major transcriptional regulator class II transactivator (CIITA). We investigated the pro-immunogenic potential of CIITA overexpression in mouse and human GB. Methods: The intracerebral growth of wildtype GL261-WT cells was assessed following contralateral injection of GL261-CIITA cells or flank injections with GL261-WT or GL261-CIITA cells. Splenocytes obtained from mice implanted intracerebrally with GL261-WT, GL261-CIITA cells or phosphate buffered saline (PBS) were transferred to other mice and subsequently implanted intracerebrally with GL261-WT. Human GB cells and (syngeneic) GB-infiltrating immune cells were isolated from surgical samples and co-cultured with GB cells expressing CIITA or not, followed by RT-qPCR assessment of the expression of key immune regulators. Results: Intracerebral vaccination of GL261-CIITA significantly reduced the subsequent growth of GL261-WT cells implanted contralaterally. Vaccination with GL261-WT or -CIITA subcutaneously, however, equivalently retarded the intracerebral growth of GL261 cells. Adoptive cell transfer experiments showed a similar antitumor potential of lymphocytes harvested from mice implanted intracerebrally with GL261-WT or -CIITA. Human GB-infiltrating myeloid cells and lymphocytes were not activated when cultured with CIITA-expressing GB cells. Tumor-infiltrating NK cells remained mostly inactivated when in co-culture with GB cells, regardless of CIITA. Conclusion: these results question the therapeutic potential of CIITA-mediated immunotherapy in glioblastoma.

## 1. Introduction

Glioblastoma (GB), WHO grade IV glioma, is the most aggressive primary brain tumor [1]. Patients with GB have a poor prognosis, despite treatment consisting of maximum surgical removal and adjuvant (chemo)radiotherapy. Patients’ 5-year survival is less than 10%, and median survival with maximum treatment is reported between 16–20 months [2,3,4]. However, median survival of only 7.9 months was also reported despite treatment, highlighting the dramatic course of the disease [5]. In contrast to other solid tumors, targeted therapies and immunotherapies have so far failed to improve the clinical prognosis in patients with GB [6,7,8,9,10].

The microenvironment of GB is shaped by the expression of immunosuppressive cytokines such as TGF-beta, interleukin (IL)-6, -10 and prostaglandin-E [11]. In this tumor microenvironment, tumor-associated macrophages (TAM) develop a pro-tumor (anti-inflammatory) phenotype and regulatory T cells dominate, while activated or cytotoxic tumor-infiltrating T-lymphocytes (TIL) are lacking [11,12,13,14,15,16,17,18,19]. GB cells express a variety of immune checkpoints (such as PD-L1 and TIGIT) and present a lower mutational load than immunotherapy-responsive tumors like melanoma and non-small-cell lung cancer [10,20,21,22].

GB cells also downregulate their expression of Major Histocompatibility Complex (MHC) type I and have low MHC-II expression, which further hampers their recognition by tumor-infiltrating T-lymphocytes (TIL; [10,11,12,23,24]). The lack of MHC-II expression in GB cells and other tumor cell lines is caused by the methylation of (one of the) promotor regions of class II transactivator (CIITA) or a lack of histone acetylation of the CIITA region [25,26,27,28]. Expression of MHC-II in tumor cells can be restored by interferon gamma treatment [29] or forced overexpression of CIITA, the major regulator of MHC-II [23]. Tumor cells that re-express MHC-II following these treatments can present tumor-associated antigens on their surface and act as accessory antigen-presenting cells [23,29]. In lung and pancreas cancers, such CIITA/MHC-II-expressing accessory antigen presenting cells can lead to tumor rejection in animal models [23,27,29,30,31]. Recently, a group [23] described the rejection of GL261 cells expressing CIITA after intracranial injection in syngeneic immunocompetent C57B/6 mice, accompanied by an abundant and rapid CD4 and CD8 lymphocyte invasion [32]. They also observed that a vaccination of C57BL/6 mice with CIITA-expressing GL261 cells prevented the subsequent growth of a new challenge of these mice with wildtype GL261 cells. They suggested that similar to its effect in lung and pancreas tumors, CIITA restoration could act as a therapeutic vaccination strategy in GB. However, murine GL261 tumor cells are also known to elicit an immune reaction against subsequent challenges in C57BL/6 mice independent of CIITA transactivation [33], and additional controls are required prior to drawing such a conclusion.

In this paper, we further investigate the extent of protection truly conferred by CIITA in GB cells in the C57BL/6 mice model and assess its effects in co-cultures of (syngeneic primary) human GB cells and tumor-infiltrating human immune cells.

## 2. Materials and Methods

### 2.1. In Vitro Experiments

#### 2.1.1. Glioblastoma Cells Lines: Transfection and Irradiation

Full-length human CIITA1-1130 was cloned in the EcoRI site of a pcDNA3.1 vector [34]. The integration of the CIITA-encoding sequence in the plasmid was checked by sequencing. Using FuGENE™ HD Transfection Reagent (PromegaTM Catalog number: E2312), GL261 (murine glioblastoma) cells were transfected with the pCIITA plasmid (GL261-CIITA). Selection of transfected cells was performed using 1 mg/mL G418 antibiotics (Sigma-Aldrich, St. Louis, MO, USA). MHC-II positive cells were purified by cell sorting (MoFlow, DakoCytomation, Milan, Italy) with the IA/IE antibody (eBioscience™) and CIITA expression was confirmed by Western Blot analysis using the CIITA 7-1H antibody (Santa Cruz Biotechnology [SCB], Dallas, TX, USA). As a control, GL261 cells were transfected with an empty pcDNA3.1 vector (GL261-WT). For some experiments, exponentially growing GL261 (-wildtype or -CIITA) cells were pre-irradiated with 10 Gy in a Gammacell^®^ 40 Exactor irradiator as described previously, keeping the cells alive but limiting their proliferation rate [35]. U87, GM2 and GM3 malignant human glioma cell lines were further transfected with the pCIITA plasmid. GM2 and GM3 cells were described in earlier publications as primary cultures [35,36]. As a result of multiple passages, these cells are now considered cell lines. After selection, cells were cultured in DMEM/F-12 medium (ThermoFisher scientific, Waltham, MA, USA) supplemented with 10% Fetal Bovine Serum (FBS Gibco™, of ThermoFisher scientific, Waltham, MA, USA) and 0.5 mg/mL G418 (Gibco™) for selection pressure and maintained in a humidified incubator at 37 degrees and 5% CO_2_ atmosphere.

#### 2.1.2. Human Primary Cell Lines and Transfection

Primary glioblastoma cultures were obtained from surgical biopsies of patients undergoing surgery for a glioblastoma resection. Informed consent was signed by the patient (16-342/16-340 reviewed by the Medical Ethical Committee of the University Medical Center Utrecht, the Netherlands). The final histopathological diagnosis of GB was made by a dedicated neuropathologist. Glioblastoma tumor pieces were cultured in a T75 flask using DMEM/F-12 (Gibco™), supplemented with 10% Fetal Calf serum (FCS, Gibco™) with penicillin (100 U/mL) and streptomycin (100 µg/mL, Penicillin-streptomycin Gibco™). Cells were transfected using Lipofectamine 2000 (Thermo Fisher) with a 1:2.5 DNA to lipofectamine ratio. 

#### 2.1.3. Immunohistochemistry for CIITA and HLA-DR Expression Analysis

CIITA and HLA-DRa expression was scored on a tissue micro array (TMA) of glioblastoma samples isolated from patients that had surgery between 2005 and 2008. All were blinded to patient survival, and expression was scored as the mean of (an average of) 3 TMA slots. Expression was scored in percentage of positive cells (0 = 0%, 1 = 1–24%, 2 = 25–50%, 3 = 51–75%, 4 = higher than 75%). Representative images were taken, showing 0% and high (51–75%) CIITA expression in the samples. 

#### 2.1.4. CD3+ (T Lymphocyte), CD11b+ (Tumor-Associated Myeloid) and CD56+ (NK-Cell) Isolation, Expansion and Co-Culture Protocol

Biopsies from patients undergoing surgery for suspected high-grade glioma were used for isolation of T-lymphocytes, tumor-associated myeloid cells and NK cells using CD3, CD11b and CD56 MicroBeads, respectively, and following the manufacturer’s instructions (MACS, Miltenyi Biotec, Bergisch Gladbach, North Rhine-Westphalia, Germany). For these experiments, GB and immune cells of 17 patients were used: 2 for TIL experiments, 10 for TAM experiments and 5 for NK-cell experiments. Briefly, under sterile conditions, biopsies were mechanically dissociated using scalpels in 200 μL GKN/BSA buffer [37] for 1 min. Then, cells were chemically dissociated in GKN/BSA buffer with collagenase type I (3700 U/mL, Worthington Biochemical) and DNase (Sigma-Aldrich, St. Louis, MO, USA) in a shaking 37 degrees incubator until the tumor pieces were (almost) dissolved for a maximum of 60 min. After red blood cell removal (1X RBC lysis buffer, eBioscience), cells were incubated for 20 min using 20 μL MicroBeads per 1 × 10^7^ cells in MACS buffer (following MACS manufacturers’ instructions, Miltenyi Biotec). After magnetic cell sorting, the positive fraction (containing CD3, CD11b or CD56-positive cells) were counted and plated in a 48 (CD3, CD56+) or a 6-well (CD11b+) plate. TAM co-culture experiments: freshly isolated CD11b+ cells were plated. One day later, U87 (-WT or -CIITA) were plated in a 1:3 TAM to tumor ratio. As a control, TAM were cultured alone. After 5 days, cells were re-isolated with CD11b microbeads as described earlier. NK-cell expansion and co-culture experiments: The floating fraction of the CD56+ cells were transferred to a new plate one day after isolation, and CD56+ cells were expanded by IL-2 and IL-15 stimulation (Gibco and Peprotech, [38]). Co-culture of CD56+ cells was performed with U87 or GM2 cells (-CIITA or -WT), and cells were cultured in a 1:4 tumor:NK-cell ratio. After 72 h, cells were separated using CD56 Microbeads and isolated for RNA analysis. For a caspase 3/7 apoptotic cell assay, we used the same set-up, and at time points 24, 48 and 72 h, CellEvent™ Caspase-3/7 Green Detection Reagent was added. The medium (also containing floating NK cells) was washed off and the remaining glioblastoma cells were fixed at the appointed time points, and the wells were stained for DAPI (Invitrogen, of ThermoFisher scientific, Waltham, MA, USA), 1:1000) and imaged using an Axioscope A1 (Zeiss, Oberkochen, Germany) fluorescent microscope. TIL activation, expansion and co-culture experiments: For expansion of CD3+ cells, a Rapid Expansion Protocol (REP) provided by the Kuball lab (University Medical Center Utrecht) was used. In short, an RPMI medium supplemented with 2.5% human serum, 100 U/mL penicillin, 100 μg/mL streptomycin was used and was enriched with irradiated PBMC (Peripheral Blood Mononuclear cells) and irradiated LCL EBC transformed B cells supplemented with OKT3 30 ng/mL (for the first 3–4 days) and IL2 50 U/m. For TIL-co-culture experiments, a syngeneic model was built using both GB cells and TIL (CD3+) derived from the same patients (PGB1 and PGB2, Appendix A). The primary tumor cells were plated in a 48-well plate. After 1 day, the cells were transfected with CIITA or an empty vector in a 1:2.5 DNA to lipofectamine ratio. After 4 h, the medium was refreshed and expanded T lymphocytes were added in a 1:5 target:effector ratio. The cells were co-cultured for 48 h and then collected and used for RNA analysis. 

#### 2.1.5. RNA Isolation, cDNA Synthesis and qPCR Analysis

Cells from co-culture experiments were lysed in QIAzol^®^ Lysis Reagent (Qiagen, Hilden, Germany) and chloroform and precipitated using isopropranol. Using QuantiTect Reverse Transcription Kit (Qiagen), cDNA was synthesized. RT-qPCR was performed using 6 ng cDNA per μL and SYBR green master mix (MERCK, Rahway, NJ, USA). Data were analyzed using the Pfaffl method, analyzing (relative) gene expression levels compared to control housekeeping genes. Housekeeping genes were validated for each separate cell type and experiment. The *y*-axis in the qPCR graphs in the figures depicts the relative gene expression. See Appendix A for a list of primers used in this study. 

### 2.2. In Vivo Experiments

#### 2.2.1. Mice 

Five-to-seven-week-old immunocompetent wildtype C57Bl/6 mice (obtained from Charles River Laboratories, Italia SRL, Calco, Italy) were used for different experiments. The mice were housed at the animal facility of the University Hospital of Liege and processed with the consent of the animal ethical committee of the University of Liège. All animals were cared for in compliance with the guidelines of the Belgium Ministry of Agriculture and in accord with the ethical committee’s laboratory animal care and use regulation (Directive 2010/63/EU of the European Parliament and of the Council of 22 September 2010 on the protection of animals used for scientific purposes). 

#### 2.2.2. In Situ Syngeneic GB Xenograft Model

For our in vivo experiments, we used the glioma mouse model with GL261 murine glioma cells. These cells were generated by injecting 3-methylcholantrene into the brains of C57Bl/6 mice, finally creating a glioma-like murine cell line. For GB research, GL261 cells are injected in (the brains of) C57Bl/6 mice to create a syngeneic mouse model, which largely prevents MHC-dependent mismatch effects, obscuring results of studies. However, immunogenicity of GL261 is reported, although the exact mechanism is unknown [33]. Fifty thousand GL261-wildtype or GL261-CIITA cells suspended in 2 μL phosphate buffered saline (PBS, Gibco) were injected intracranially (i.c.) in the right striatum of the mice using a stereotactic frame (coordinates with respect to the bregma: ML-2.5, AP-1, DV-3) as described previously [35]. For tumor growth experiments, the mice were anesthetized at day 21 post-injection and tumor volume was assessed with gadolinium-based contrast using 9.4 T Magnetic Resonance Imaging T1 sequence and a finger imaging coil. After imaging, the mice were killed by cervical dislocation and their brains were isolated for immunohistochemistry.

#### 2.2.3. Local Vaccination

A total of 50,000 GL261-CIITA cells in 2 μL PBS were injected in the left striatum; PBS was injected as a control-sham vaccination. Three weeks after, 50,000 GL261-WT cells were injected in the right (contralateral) striatum of the same mice. After 3 weeks, the mice were euthanized for tumor volume assessment. 

#### 2.2.4. Subcutaneous Vaccination

A total of 50,000 pre-irradiated GL261-WT cells, -CIITA cells or PBS-only as control (volume 2 μL) were injected subcutaneously in the flank of C57Bl/6 mice. After 21 days, these mice were challenged with an intracranial injection of GL261-WT cells. The mice were euthanized 21 days later by cervical dislocation after sedation. The tumors were harvested for histology and maximum tumor diameter was measured on scanned hematoxylin and eosin-stained slices. 

#### 2.2.5. Adoptive Cell Transfer (ACT) Experiments

Spleens were harvested from C57Bl/6 mice that were injected 21 days prior with 2 μL PBS with 50,000 GL261-WT, -CIITA or only PBS. Splenocytes were dissociated mechanically, and red blood cells were discarded using ACK lysing Buffer (Gibco). One day after intracranial injection with GL261-WT cells, the splenocytes were injected intraperitoneally in a 1:100 tumor cell–splenocyte ratio. As a control, “naïve” splenocytes from mice that were injected intracranially with sterile PBS were used. Survival was assessed up to 63 days (n = 42 mice per group). 

### 2.3. RNA Sequencing Data

For RNA seq profiler experiments, cell pellets from GM1, GM2, GM3 and GL261 cells (-CIITA or -WT) were collected from 80% confluent T75 flasks and processed for RNA extraction using the Qiazol lysis reagent, followed by chloroform precipitation. For analyzing the integrity of the RNA, the Agilent 2100 bioanalyzer (Agilent Technologies, Palo Alto, CA, USA) was employed with the Agilent RNA 6000 Nano Kit (Agilent technologies). A total of 1000 ng RNA per sample was used as input. Library preparation was performed with the Illumina Truseq^®^ mRNA HT kit (Illumina Inc., San Diego, CA, USA). Purified libraries were quantified by qPCR with KAPA Library Quantification Kits (Kapa Biosystems, Wilmington, MA, USA). Sequencing was then performed using Illumina’s NextSeq 500 system, run for 75 cycles.

### 2.4. Statistics

Statistics were performed using R studio (R version 3.6.0). Graphs and survival curves were made using R studio or GraphPad (version 8). For non-parametric data, the Mann–Whitney test was applied. For normal distributed data, a non-paired *t*-test was applied.

## 3. Results

### 3.1. Forced Expression of CIITA in Murine GL261 Cells

#### 3.1.1. Tumor Growth and Animal Survival following Intracerebral Injection of CIITA-Expressing GL261 Cells in Syngeneic C57Bl/6 Mice

Forced CIITA expression in GL261 cells following plasmid transfection (GL261-CIITA) restored their expression of MHC-II (Appendix A). Syngeneic C57Bl/6 mice injected with GL261-CIITA intracerebrally did not develop any intracerebral tumor after 3 weeks, while the injection of GL261 cells transfected with an empty vector (pcDNA 3.1 plasmid, GL261-WT) resulted in large intracerebral tumors (87.95 mm^3^ [SD 17.75] vs. 0 mm^3^, *p* < 0.0001, Figure 1a–c). 

In follow-up experiments, the survival of mice injected with GL261-WT or -CIITA was assessed up to 63 days (n = 10 per group). Median survival after intracerebral injection with GL261-WT was 35 days, whereas only one mouse of the GL261-CIITA group died after 47 days, and the rest survived up to 63 days (Hazard ratio [HR] 6.944 [95% confidence interval [CI] 1.491 to 32.33], *p* = 0.0114, Log Rank, Figure 1d).

#### 3.1.2. Intrastriatal Injection of CIITA-Expressing GL261 Cells Protects against a Subsequent Contralateral Wildtype Tumor Challenge

Twenty-one days after injection with PBS (sham-vaccination) or GL261-CIITA cells (CIITA-tumor vaccination) in the left hemisphere of C57Bl/6 mice, the mice were challenged with GL261-WT cells in the contralateral hemisphere. While sham-vaccinated (PBS) mice all developed large tumors, GL261-CIITA vaccinated mice either fully rejected the subsequent wildtype tumors (40%) or only grew very small tumors (n = 5, 1.08 mm^3^ [SD 1.88] versus n = 8, 74 mm^3^. [SD 27.5], *p* = 0.0001, Figure 1e–g). Due to their rapid growth in the brain of C57Bl/6 mice, we could not, however, assess the effect of a vaccination with GL261-WT cells on the subsequent growth of a second challenge with such wildtype cells. 

#### 3.1.3. Subcutaneous Vaccination with GL261-CIITA or GL261-WT Cells Leads to Intracerebral Tumor Rejection in C57Bl/6 Mice

Previous reports have shown that GL261 cells are immunogenic, despite being syngeneic in origin to C57Bl/6 mice [33]. To assess whether the intracerebral rejection of GL261-WT following a challenge with GL261-CIITA was due to their actual CIITA expression in the previous experiment, C57Bl/6 mice were injected subcutaneously (sc.) in the flank with either PBS, GL261-WT or GL261-CIITA cells. After 2 weeks, GL261-WT cells were injected in the right striatum of mice and the growth of the tumors was assessed by immunohistochemistry on day 14 (Figure 2a). On day 14, histological analysis showed that three-fourths of PBS-primed mice had developed large brain tumors (mean diameter 2585.25 μm [SD 2723.7]), while only one-sixth of the GL261-WT and one-fourth of the GL261-CIITA-primed mice developed a tumor (mean diameter 158.67 μm [SD 388.7] and mean diameter 39.8 μm [SD 79.5]; one-way ANOVA *p* = 0.045, Figure 2b,c).

#### 3.1.4. Adoptive Cell Transfer (ACT): Vaccination with Splenocytes of Intracerebrally Primed Mice

To assess the potential of splenocytes primed with CIITA-expressing GB cells to suppress the intracerebral tumor growth of wildtype GB cells, we performed adoptive cell transfer experiments (Figure 2d). There was no difference in survival between mice injected with splenocytes primed with either GL261-WT or –CIITA (n = 42 per group, HR 1.008 [95% CI 0.57 to 1.796], *p* = 0.978, Log Rank). However, both groups tended to survive longer than mice injected with PBS-primed splenocytes (HR 1.494 [95%CI 0.857 to 2.604, Log Rank *p* = 0.14 [PBS versus WT] and HR 1.519 [95% CI 0.871 to 2.649], *p* = 0.15 [PBS versus CIITA], Figure 2e). More mice injected with GL261-WT or -CIITA-primed splenocytes survived until termination of the experiment (19/42 in both groups, day 63) as compared to mice injected with non-primed splenocytes (11/42, *p* = 0.0685 for both groups, Chi-square test, Figure 2e). 

### 3.2. Forced CIITA Expression in Human Malignant Glioma Cell Lines

Using a CIITA-encoding plasmid, the human malignant glioma U87 cell lines and our proprietary human glioblastoma cell line GM2 were transfected, resulting in HLA-DR overexpression in both cell types. Primary GB cells also expressed CIITA after transfection (Appendix A). These malignant glioma cells and their empty-pcDNA vector-transfected (pcDNA3.1) wildtype counterparts (-WT) were co-cultured with tumor-associated myeloid cells (TAM) and NK cells (obtained from surgical samples of GB surgeries, as described in the material and methods section).

#### 3.2.1. RNA Sequencing of WT and CIITA-Expressing Human Malignant Glioma Cell Lines

Whole exome RNA sequencing confirmed an increased expression of both HLA-I and –II transcripts of the U87 and GM2-CIITA cell lines as compared to WT counterparts (Appendix A). Gene ontology analysis of the global results did not reveal any alteration of the pathways involved in immunity. Likewise, at the messenger RNA level, analyses did not reveal any differential expression of important immune modulator genes, including, among others, IL-6, PD-L1 and TGF-beta. However, RAB11FIP1, a gene associated with M1-differentiation in tumor-(glioma)-associated macrophages (TAM), was upregulated (mean log fold change 4.6 [U87 8.6, GM2 2.3 and GM3 3.0]). 

#### 3.2.2. TAM Develop a Pro-Tumor Phenotype When Co-Cultured with U87 Cells, Regardless of Their CIITA Expression Level

Phenotypically, TAM similarly structured the growth of both WT and CIITA-expressing U87 cells in vitro as compared to U87 cells grown in monoculture (Figure 3a). HLA-DR was upregulated in TAM co-cultured with U87-CIITA compared to U87-WT (*p* = 0.034), while TNF-alpha, IL6 and IL1b remained unchanged. TAMs expressed more CD163 when co-cultured with tumor cells, regardless of CIITA expression, than when grown in monoculture (*p* = 0.01). A similar trend was observed for CD206 and TGF-beta expression, although this did not reach significance (both *p* = 0.063, Figure 3b). Other genes associated with tumor-associated myeloid cells in glioma were tested but there were no significant changes (Appendix A). 

#### 3.2.3. Effects of CIITA-Expressing Glioblastoma Cells on Tumor-Infiltrating NK (CD56+) Cells

NK cells isolated from glioblastoma surgical samples were co-cultured with U87 (Figure 3(c1)) or GM2 cells (Figure 3(c2)) with or without CIITA overexpression. LAMP1 expression in NK cells—a potential marker of anti-tumor activity—was consistently high, independent of CIITA expression in U87 or GM2. However, the expression of CD69 and granzyme B—important effectors of NK-cell function—remained low in NK cells in both conditions. There was a two-fold increase of CD137 expression in NK cells after 72 h of co-culture with both GM2- and U87-CIITA with respect to NK cells co-cultured with wildtype tumor cells (*p* = 0.11 and *p* = 0.23). Likewise, the expression of the CD226 co-activator increased steadily but not significantly up to 72 h in NK cells co-cultured with U87-CIITA) and only transiently (at 48 h) in co-culture with GM2-CIITA (Appendix A). 

In a caspase 3/7 apoptotic cell assay following co-culture of NK cells with GB cells expressing CIITA or not, we reproducibly measured a higher percentage of apoptotic GB cells after 72 h of co-culture with NK cells than after 24 h of co-culture. This increase was, however, not influenced by CIITA overexpression in the glioblastoma cells (mean 5.43% versus 2.3% [Wildtype] and mean 3.99% versus 2.2% [CIITA], n = 2, Appendix A).

#### 3.2.4. Syngeneic Co-Cultures of Tumor-Infiltrating Lymphocytes and Primary Human Glioblastoma Cells

FoxP3 expression increased in tumor-infiltrating T-lymphocytes (TIL) after co-culture with syngeneic GB-CIITA. CIITA RNA expression was successfully upregulated in the primary GB cells of two patients by plasmid transfection (PGB1 and PGB2) (S4B[a]). FoxP3 was upregulated in TIL after co-culture with CIITA-expressing syngeneic tumor cells (PGB1) as compared to co-cultures with the corresponding wildtype tumor cells (*p* = 0.04). A similar yet not statistically significant upregulation was seen in PGB2 (Figure 3(d1)). The CD155 expression level (expressed by the PGB cells) was similar between CIITA-expressing and PGB-WT cells in these co-cultures (Figure 3(d2)). The expression of its co-inhibitory and co-stimulatory receptors (respectively, TIGIT and CD226) on TIL did not change. Additionally, we did not observe any difference in the expression of other checkpoint inhibitors such as PD-1 and CTLA4 (Figure 3(d1)). 

### 3.3. CIITA Expression in Glioblastoma Correlates with HLA-DR Expression, but Does Not Determine Survival in Patients

CIITA expression was assessed in 76 glioblastoma samples using immunohistochemistry (Figure 4c). Most samples showed no CIITA expression (expression score [ES] 0, n = 57 (75%), Figure 4c), and in 13 cases (17%) fewer than 25% of the cells in the TMA spot showed CIITA expression (ES 1). Only six samples showed CIITA expression in more than 25% of the cells (ES 2 or higher, Figure 4c). Survival between patients with high (≥25%) or low (<25%) expression did not differ (HR 0,6736 [95% CI 0.3344 to 1.357], *p* = 0.267, Log Rank, Figure 4a), and this did not change when analyzing 0% expression versus >0% expression (HR 0.8221 [95% CI 0.49 to 1.371], *p* = 0.453, Log Rank, Appendix A). HLA-DRa expression was analyzed in 104 samples using the same method. Most samples showed an expression level below 25% (n = 72, 69%), with six cases showing no expression. In 30.5% of the samples, expression was between 25% and 100%. There was a positive correlation between HLA-DRa and CIITA expression (R = 0.26, *p* = 0.0152, Figure 4d). Survival did not differ between high (≥25%) or low (<25%) expression of HLA-DR (1.271 [95% CI 0.81 to 2.01], *p* = 0.881, Log Rank, Figure 4b)).

Performing the same analysis in the TCGA database confirmed these results (median split, n = 80 with high CIITA expression; n = 81 with low CIITA expression; n = 81 per group for HLA-DR). Higher CIITA expression was positively correlated with higher HLA-DRA expression (R-value = 0.63, *p* < 0.0001, Figure 4g). Survival did not improve with high CIITA or HLA-DRa expression (*p* = 0.39 [HR 1.2] and *p* = 0.31 [HR1.2], Figure 4e,f—patients from TCGA database, using GEPIA software, accessed on 01-10-2023). Comparing the highest versus the lowest 25th percentiles and the highest versus lowest 10th percentiles instead of the median split did not alter this finding (*p* = 0.098 and *p* = 0.58, Appendix A).

## 4. Discussion

CIITA-based anti-GB treatment can restore antigen presentation on MHC class II molecules at the surface of GB cells, and it is thought to induce an immune-mediated anti-tumor immune response [23,32]. In vivo experiments showed that murine GL261-CIITA injected in the striatum of syngeneic C57Bl/6 mice did not grow and elicited a strong CD4+ and CD8+ lymphocytic reaction, whereas large tumors emerged after similar injections of GL261-WT cells [32]. A vaccination strategy, performed by a prior injection of GL261-CIITA in the contralateral hemisphere, showed tumor growth retardation after injection of GL261-WT in the other hemisphere. This suggested that forced expression of CIITA in tumor cells could induce a potent antitumor response against glioblastoma. However, these experiments did not control for the potential effect of a vaccination by GL261-WT cells due to the rapid growth of these cells and the early death of most animals injected with such cells intracranially. In the present paper, we could reproduce these results, but we also performed a subcutaneous vaccination of C57Bl/6 mice with WT or CIITA-expressing cells. Our control showed a similar protection of GL261-WT and GL261-CIITA cells against a subsequent intracerebral challenge with GL261-WT. These findings are reminiscent of earlier works, where experiments showed that a subcutaneous vaccination with GL261-WT cells more than 7 days prior to intracranial injection of GL261 cells led to tumor rejection or growth retardation [33,39,40]. This seems due to an incomplete syngenicity between the GL261 cells and their C57Bl/6 hosts that results in the development of a ’spontaneous’ immune reaction against GL261 cells. It is unknown why the immunogenicity of GL261 cells in C57Bl/6 mice occurs. Of note, ~40% of the mice implanted intracerebrally with WT cells appeared to be cured in our long-term survival experiments, suggesting that their tumors could be cleared ‘spontaneously’ by their hosts (see Figure 1d). While this immunogenicity of GL261-WT cells has not prevented studying large, relevant immune mechanisms like checkpoint inhibition using this model in the past [41], it certainly could mask smaller immunomodulatory effects. We thus performed additional adoptive cell transfer (ACT) experiments with large numbers of animals (n = 42 per group), using lymphocytes from mice primed intracerebrally with either GL261-WT or -CIITA cells, or sham-operated. These ACT experiments showed an identical (almost significant: *p* = 0.0685) protective effect of the cells primed with either GL261-WT or -CIITA-expressing cells with respect to the transfer of cells from sham-operated mice. Together, these experiments do not plead for any significant enhancement of the immune response against brain tumors by the restoration of CIITA expression in GB in this murine model.

The contrast between the intracerebral growth of CIITA-expressing and GL261-WT cells could be due to mechanisms other than a pure immunologic rejection of the tumors. For instance, a slower pace of growth of these cells (through multiclonal selection or effects of CIITA) could provide sufficient time for the mice to then develop their immune reaction against the tumor cells, while mice injected with fast-growing WT cells would more often die from tumor growth prior to the development of a sufficient immune reaction. A thorough study of the potential effects of CIITA on the pace of growth of glioma cells might be relevant in this view, but it is beyond the scope of the present study. 

Given the limitations of our murine model, we assessed the effects of CIITA expression in tumor cells on human immune cells. We isolated immune cells from surgical samples of GB to select immune cells that express homing factors that direct them towards the tumor [42,43]. For the study of the interaction between GB and CD3+ T lymphocytes, which work in an HLA-restricted fashion, co-cultures of primary cells from surgical donors with lymphocytes derived from the same tumor (syngeneic match) were performed. For non-HLA restricted TAM and NK-cell co-culture experiments, GB cell lines were used, given their homogeneity and ease of culture.

In our syngeneic co-cultures of primary human GB and their infiltrating lymphocytes, the restoration of CIITA expression in GB cells did not reduce the expression of the checkpoint inhibitors PD1, TIGIT, CD226 or CTLA4. It even led to an increased expression of FoxP3, a marker of regulatory TIL. This latter finding is in accordance with a previous report in murine tumors in vivo [32] and suggests that CIITA expression by GB cells could result in negative immunomodulatory effects. 

Co-culture tumor-infiltrating myeloid (TAM) cells with GB-CIITA increased the MHC-II expression in TAM as compared to a co-culture with GB-WT cells. Although this could theoretically help antigen presentation to T lymphocytes in the tumor and help at inducing an anti-tumor reaction, expression of CIITA by the tumor cells did not affect the overall tumor-supportive phenotype of myeloid cells (in terms of gene expression of both interleukins and receptors) and their pro-tumor biological effect in culture.

NK cells represent only a small percentage of the tumor microenvironment of GB [44], but they have proven activity against these tumors [45]. At baseline, we observed only a low expression of NK-cell effectors granzyme B and CD69 in our NK cells. This may be due to the IL-15 treatment needed for their amplification in vitro [38,46] or to long-lasting inhibitory effects of their previous interactions with GB in vivo [47]. The co-culture of these NK cells with CIITA-expressing GB cells did not increase their expression of granzyme B and CD69, but reproducibly increased that of CD137 co-activator and of CD226 activator at their surface, thus showing some, although small, potential pro-immunologic effects of CIITA [48]. This potential pro-immunogenic effect of CIITA did not, however, result in any additional GB cell-killing by NK cells in our co-culture experiments.

Altogether, the results of our co-cultures of CIITA-expressing human GB cells with different subsets of tumor-infiltrating immune cells do not suggest any major immunomodulatory effect of CIITA on the tumor microenvironment. This is supported by our RNA sequencing results obtained on human GB cells following CIITA overexpression, which did not show any significant alteration of the expression of interleukins, TGF-beta or immunity-related pathways. Likewise, the analyses of a proprietary series of patients and of the TCGA gene expression database do not show any correlation between patient survival and CIITA expression in GB.

## 5. Conclusions

Despite previous reports of the potential of CIITA gene therapy in brain tumors to induce a potent antitumor immune response, additional control experiments in the same GL261/C57Bl/6 glioma mouse model strongly mitigate the importance of this mechanism. Moreover, the experiments performed on human GB cell lines and syngeneic primary GB cultures and infiltrating immune cells show few pro-immunogenic, and even immune-suppressing, effects of CIITA re-expression in GB cells and question the potential of such a strategy for the treatment of human GB.

## Figures and Tables

**Figure 1 cancers-16-00193-f001:**
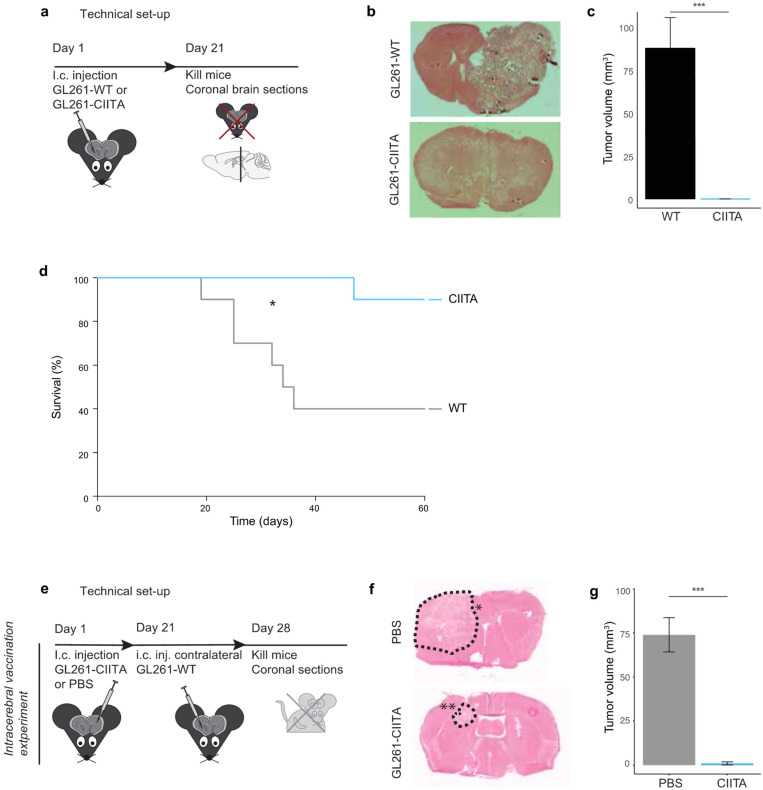
(**a**) Set up of experiment. (**b**) Coronal brain slices of mice after the experiment. (**c**) Direct intracranial injection of GL261-wildtype or -CIITA cells in C57Bl/6 mice. The bargraph represents the mean ± SD. (**d**) Survival curve of mice after direct i.c. injection with GL261-WT or -CIITA. Survival was significantly longer after injection with GL261-CIITA. (**e**) Set up of intracerebral contralateral vaccination experiment. (**f**) Coronal brain slices of mice after the vaccination experiment. The dashed line depicts the tumor engraftment after (*) PBS or (**) GL261-CIITA vaccination in the contralateral hemisphere. (**g**) The bargraph represents the mean ± SD after GL261-CIITA vaccination PBS vaccination. **Abbreviations:** I.c. = intracranial. WT = wildtype. ACT = Adoptive Cell Transfer. PBS = Phosphate Buffered Saline. * *p* < 0.05. *** *p* < 0.0001.

**Figure 2 cancers-16-00193-f002:**
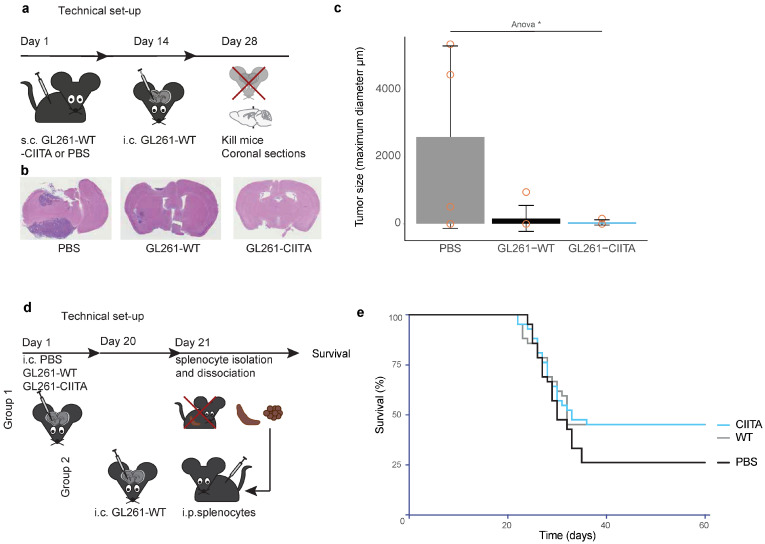
(**a**) Technical set up of peripheral vaccination experiment. (**b**) Representative images of coronal brain slices. (**c**) Bargraph, representing mean, SD and separate datapoints (orange). ANOVA *p* = 0.045. (**d**) Set-up of peripheral vaccination experiment. (**e**) Survival curve, representing survival of mice injected intracerebrally with GL261-WT and subsequently injected with splenocytes that were primed with either GL261-WT, -CIITA or PBS (sham-vaccinated). Abbreviations: S.c. = subcutanetous (injection). I.c. = intracranial (injection). WT = wildtype. PBS = Phosphate Buffered Saline. m = micrometer. * *p* < 0.05.

**Figure 3 cancers-16-00193-f003:**
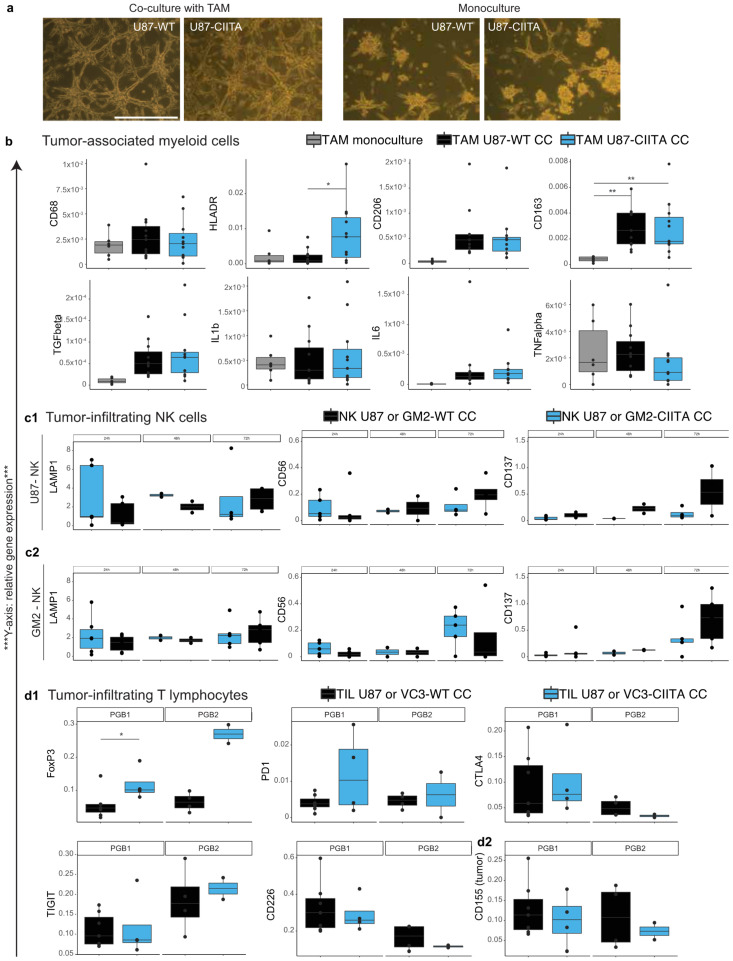
(**a**) Phenotypic changes after co-culturing tumor-associated macrophages with U87-CIITA or -wildtype cells versus U87 monocultures. Scale bar in white: 100 micrometer, the same scale for all images. (**b**–**d**) Gene expression of the indicated genes was determined by qPCR. Each bar represents the mean ± SD relative gene expression. * *p* < 0.05, ** *p* = 0.01. (**b**) Gene expression in tumor-associated macrotphages after co-culture with U87 glioblastoma cells and in monoculture. (**c**) Gene expression in tumor-infiltrating NK-cells after co-culture with U87 (**c1**) and GM2 (**c2**). (**d**) Gene expression of genes expressed in tumor-infiltrating T lymphocytes (**d1**) and of CD155 expressed in tumor cells (**d2**). **Abbreviations:** TAM = tumor-associated myeloid cells. NK = Natural Killer cell. TIL = Tumor-infiltrating T lymphocytes. WT = wildtype. CC = co-cultured.

**Figure 4 cancers-16-00193-f004:**
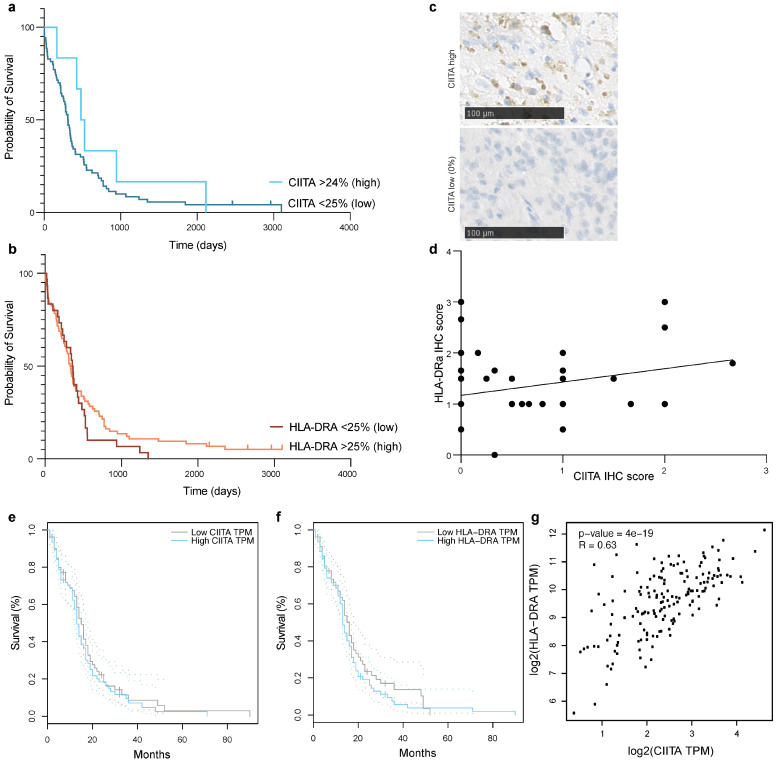
(**a**) Survival curve of patients with glioblastoma with low (<25%) or high (>24%) CIITA or (**b**) HLA-DRa expression. (**c**) Two representative images of TMA with glioblastoma cells with high or no CIITA-expression. (**d**) Correlation between HLA-DRa expression and CIITA expression in glioblastoma. Using Gepia.cancer.pku online software, we calculated Kaplan meier curves for CIITA (**e**) and HLA-DRa (**f**) expression in patients with glioblastoma of TGCA cancer atlas. (**g**) Correlation between HLA-DRa and CIITA expression in glioblastoma.

## Data Availability

All original data are readily available upon motivated request by contacting the corresponding author.

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
