# Peer review of "Limited Effects of Class II Transactivator-Based Immunotherapy in Murine and Human Glioblastoma"

_cancers, 2023, doi:10.3390/cancers16010193_

Round 1
Reviewer 1 Report
Comments and Suggestions for Authors
Dear authors,
this is a descriptive work about the therapeutic potential of CIITA-mediated immunotherapy in glioblastoma multiforme (GB). The text is nicely written, the experiments performed on human GB cell lines and syngeneic primary GB 440 cultures and infiltrating immune cells show little pro-immunogenic, and even immune- 441 suppressing effects of CIITA re-expression in GB cells. In my opinion, some parts of this work are valuable to understand the complex scenario of glioblastoma multiforme.
Reviewer 2 Report
Comments and Suggestions for Authors
The major histocompatibility complex type II is downregulated in glioblastoma (GB) due to the silencing of the major transcriptional regulator Class-II-Trans activator (CIITA). Previous reports suggest the potential of CIITA gene therapy in brain tumors to induce a potent antitumor immune response. The studies reported in this paper reveal however that wildtype and CIITA-expressing cells equivalently protect against subsequent intracerebral challenges with mouse GL261 cells. Human GB-infiltrating myeloid cells, lymphocytes and NK cells remained mostly inactivated and/or tumor-supportive when co-cultured with GB cells, regardless of CIITA. These results question the therapeutic potential of CIITA-mediated immunotherapy in glioblastoma.
Comments on the Quality of English LanguageEnglish reasonably good.
Reviewer 3 Report
Comments and Suggestions for Authors
The manuscript "Limited Effects of CIITA-Based Immunotherapy in Murine and Human Glioblastoma" represents an original article on the immunological regulatory mechanisms in glioblastoma, focusing on the role of the major transcriptional regulator Class_II_Transactivator (CIITA). Authors dispute the previously reported findings, namely, the previously published suggestion that CIIITA restoration could act as a therapeutic vaccination strategy in glioblastoma. In contrast, authors have presented here their own data, justifying another hypothesis: an immune reaction against GL261 cells themselves. Such a scientific dispute is of utmost value; therefore the new data should be made available to the global scientific community.
Considering the dismal prognosis of glioblastoma, the topic undoubtedly is timely and important. The contents of the manuscript correspond to the scope of the journal “Cancers”, the section “Tumor microenvironment” and special issue “Targeting the Tumor Microenvironment Volume II”. Up-to-dated technologies have been implemented for this research. The manuscript is detailed and richly illustrated. The level of English language is generally good, although few misprints should be corrected, as further indicated.
Few corrections would be recommended, please:
1. Please, provide a short characteristic of GL261 cells, as some readers might have no experience with this cell line. How this murine glioblastoma cell line has been obtained? Why is it immunogenic?
2. Clarify, please, in “Materials and methods” (subsections 2.1.2. and 2.1.4.), how many cases of human glioblastomas were included in this study.
3. The survival of glioblastoma patients can be significantly shorter than 16 – 20 months: see, please, Jakovlevs et al., 2019 (PMID: 32146793), reporting median survival of 7.9 months. Although such data might reflect lower level of medical care outside specialised/ excellence centres, they also highlight the extent of the true challenges in treating glioblastoma.
4. Although the level of English is reasonably high, few minor misprints should be corrected, e.g., use of plural vs singular (line 42); misprints (e.g., “challence”, line 70; among others); structure of the sentence (lines 257 – 258); incomplete sentences (line 372: “with”?); redundant phrases (“Click or tap here to enter text”; line 360), etc. In line 394, do you mean death of cells or of animals?
5. Please, format the list of authors and affiliations according to the Instructions for Authors, as indicated by “Cancers”.
6. Consider, please, if it is mandatory to use an abbreviation in the title?
Finally, I would like to thank the authors for their contribution. It was a pleasure and a true honour to review this manuscript.
Comments on the Quality of English LanguageAlthough the level of English is reasonably high, few minor misprints should be corrected, e.g., use of plural vs singular (line 42); misprints (e.g., “challence”, line 70; among others); structure of the sentence (lines 257 – 258); incomplete sentences (line 372: “with”?); redundant phrases (“Click or tap here to enter text”; line 360), etc.
